# LaDEEP: A Deep Learning-based Surrogate Model for Large Deformations of Elastic-Plastic Solids

## Abstract

The scientific computing for large deformations of elastic-plastic solids is critical for numerous real-world applications. Classical numerical solvers rely primarily on local discrete linear approximations, which are constrained by an inherent trade-off between accuracy and efficiency. Recently, Deep Learning models have achieved impressive progress in solving PDEs. While previous models have explored various architectures and constructed coefficient-solution mappings, they are designed for general instances without considering specific problem properties and hard to accurately handle with complex elastic-plastic solids involving contact, loading and unloading. In this work, we take stretch bending, a popular metal fabrication technique, as our case study and introduce LaDEEP, a deep learning-based surrogate model for **La**rge **De**formations of **E**lastic-**P**lastic Solids. We encode the partitioned regions of the involved solids into a token sequence to maintain their essential order property. To characterize the physical process of the solid deformation, a two-stage Transformer-based module is designed to predict the deformation with the sequence of tokens as input. Empirically, LaDEEP achieves five magnitudes faster speed than finite element methods with a comparable accuracy, and gains 20.47% relative improvement on average compared to other deep learning baselines. We have also deployed our model into a real-world industrial production system, and it has shown remarkable performance in both accuracy and efficiency.

## 1 Introduction

The scientific computing for large deformations of elastic-plastic solids (Bathe & Ozdemir, 1976) is essential in continuum mechanics, which is widely used in various areas such as civil engineering (AbouRizk & Hajjar, 1998), aerospace (Phanden et al., 2021), and nuclear materials (Allen et al., 2012). The deformation of a solid typically results from the application of external loads and constraints. Large deformations occur when the extent of deformation becomes significant enough to invalidate the assumptions of infinitesimal strain theory (Bower, 2009). Metals are particularly important subjects of study due to their distinct elastic and plastic behaviors. In the elastic regime, metals tend to return to their original shape after deformation, but only up to a certain threshold. Beyond this limit, when sufficient load is applied and the material enters the plastic state, permanent deformation occurs. Upon the release of the load, the elastic component of the deformation will recover, while the plastic deformation remains, causing the material to rebound slightly and assume its final shape. An accurate and efficient solver for such complex task is in urgent demand.

Figure 1 illustrates a practical example known as stretch bending (Clausen et al., 2000), one of the most widely used metal fabrication techniques (Murr et al., 2012). This process consists of two distinct stages: loading and unloading. During the loading phase, the metal workpiece is positioned on the machine and securely held by two working arms. These arms move and rotate, applying force to the workpiece, causing it to lengthen and curve around a mold, which acts as a constraint. In the unloading phase, the applied loads are released, allowing the metal to rebound and assume its final shape. Our primary interest lies in predicting the final shape of the workpiece given the influence of the applied loads and constraints.

This fabrication technique involves large deformation of elastic-plastic solids, governed by partial differential equations (PDEs). The workpiece corresponds to the solution domain, the mold represents the boundary condition, and the loads from the movement of the working arms are modeled as source terms. Traditional methods, such as Finite Element Methods (FEM) (Reddy, 2019), depicted in Figure 2a, rely on local discrete linear approximations. The material is subdivided into smaller

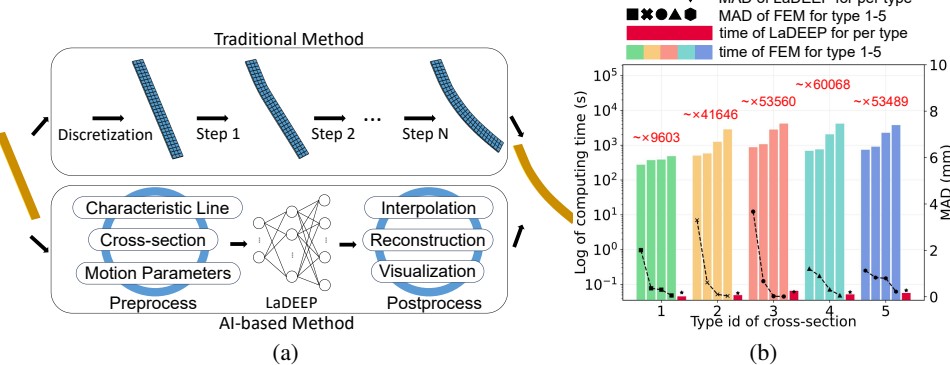

Figure 1: The stretch-bending process. It consists of two elastic-plastic stages including loading process and unloading process.

elements through meshing techniques in the solution domain (Alliez et al., 2005), and solutions are incrementally approximated. However, these methods face a trade-off between computational accuracy and efficiency due to the discretization. Higher accuracy could be achieved with finer and more regular elements, though in practical cases, a large number of irregularly shaped elements may be encountered. Moreover, discretizing continuous domains introduces tens of thousands of degrees of freedom, resulting in an exponential rise in computational complexity (Köppen, 2000).

Recently, deep learning (DL) has demonstrated significant potential in solving PDEs (Raissi et al., 2019; Lu et al., 2019; Li et al., 2022; Wen et al., 2022). Physics-Informed Neural Networks (PINNs) (Raissi et al., 2019) incorporate PDE constraints into the loss function and leverage automatic differentiation to optimize the model, effectively transforming the network into a solver for specific PDE instances. However, PINNs struggle to generalize to similar instances without retraining. Neural operators, like FNOs (Li et al., 2022; Wen et al., 2022; Rahman et al., 2022; Gupta et al., 2021; Li et al., 2020; 2023) and DeepONet (Lu et al., 2019), learn the mappings between infinite-dimensional function spaces, offering a broader applicability. Transformer-based (Vaswani et al., 2017) methods have also achieved noteworthy progress in PDE-solving (Liu et al., 2021; Li et al., 2024a; Wu et al., 2023; 2024). LSM (Wu et al., 2023) employs an attention-based hierarchical projection network that incorporates spectral methods (Gottlieb & Orszag, 1977) to reduce the high-dimensional data into a compact latent space in linear time. Similarly, Transolver (Wu et al., 2024) introduces physics-attention to adaptively partition the discretized domain into a series of learnable slices, capturing the underlying physical states. While these methods are generally effective for problems in fluid mechanics or simple solid mechanics, they tend to underperform in our problem, which involves non-smooth boundary conditions (solid contacts) and multi-stage processes (unloading and loading) due to the ignorance of characteristics of large deformation and staged modeling.

Figure 2: (**a**) Traditional Method vs AI-based Method; (**b**) Comparison between LaDEEP and FEM on the accuracy and speed, respectively.

To address these challenges, we begin by analyzing the inherent properties of the task and make a pioneering attempt to apply deep learning to the large deformations of elastic-plastic solids. Focusing on a highly relevant industrial scenario, the stretch bending, we introduce LaDEEP as a surrogate model to approximate the complex multi-stage process. The model follows an Encoder-Processor-Decoder procedure (Battaglia et al., 2018). We develop several specialized modules to encode property-aware token sequences, which represent sequentially partitioned regions corresponding to solids that retain essential order properties. Then, a two-stage Transformer-based module, the Deformation Predictor (DP), is proposed as a processor to characterize the loading and unloading process. By applying attention mechanisms to the token sequences , DP effectively captures the complex underlying interactions between objects (see Figure 5). To validate our approach, we first create a new dataset

to fill the gap of data scarcity in this field and conduct extensive experiments comparing LaDEEP with classical FEM method and other deep learning models. LaDEEP demonstrates significant performance improvement over other alternatives. More importantly, we have successfully deployed LaDEEP in practical scenario. Our main contributions are as follows.

- We are, to the best of our knowledge, the first to apply deep learning into solving complex large deformations of elastic-plastic solids. We propose LaDEEP, a novel deep learning-based framework tailored for an industrial instance, stretch bending task, with modules designed to consider its key order properties.

- We introduce the Deformation Predictor (DP), a two-stage Transformer-based module that effectively captures complex deformation behaviors through cross-attention and self-attention mechanisms.

- We generate a new dataset for our case, supporting our approach and filling a critical data gap in this domain. Experiments show that LaDEEP achieves five magnitudes faster speed than classical FEM method (Reddy, 2019) with a comparable accuracy (Figure 2b), and gains 20.47% relative improvement on average compared to deep learning baselines across all evaluation metrics.

- We complete the design of seven products in practical scenario. Two of them have already been put into real production with mean absolute distance (MAD) $0.305mm$ on average.

## RELATED WORK

**Physics-Informed Neural Networks**. These approaches formulate the PDEs, including governing equations, source items, initial conditions, and boundary conditions, as loss functions within DL models (Raissi et al., 2019). During training, the output of the model progressively conforms to the PDE constraints, ultimately providing an accurate approximation of the PDE solution. However, These approaches require the precise formula of PDEs, making them difficult to apply to real-world scenarios with incomplete observations. Additionally, they are typically limited to solving a single problem instance, and any change in parameters requires retraining the models.

**Neural Operators**. The idea of neural operators is to learn mappings between two infinite-dimensional function spaces. The most prevailing models are FNO (Li et al., 2022), which approximates integration with linear projection in the Fourier domain. Building on this, various variants (Wen et al., 2022; Rahman et al., 2022; Gupta et al., 2021; Tran et al., 2021; Li et al., 2020; 2023; 2024b; Bonev et al., 2023) have been proposed with crafted architectures to improve the accuracy, efficiency and application extensions. DeepONet (Lu et al., 2019) is also a prevalent model for operator learning which is designed based on the Universal Approximation Theorem for Operator (Chen & Chen, 1995). However, as shown in our experiments, they tend to degenerate in our case, which involves non-smooth boundary conditions (solid contacts) and multiple stages (unloading and loading) due to the lack of consideration over the properties of solids and staged modeling.

**Transformer-based PDE Solvers**. Transformers (Vaswani et al., 2017) have been employed to solve PDEs. HT-Net (Ma et al., 2022) combines Swin Transformer (Liu et al., 2021) with multigrid methods (Wesseling, 1995) to capture multiscale spatial relationships. FactFormer (Li et al., 2024a) enhances efficiency by leveraging a low-rank structure with multidimensional factorized attention. LSM (Wu et al., 2023) is introduced to address the high-dimensional complexity of PDEs by utilizing spectral methods (Gottlieb & Orszag, 1977) within a learned latent space. Transolver (Wu et al., 2024) introduces physics-attention to dynamically partition the discretized domain into learnable slices that capture the underlying physical states. However, these approaches struggle to handle our case due to the lack of the inherent property of the large deformations of elastic-plastic solids.

## 2 METHOD

The overview of LaDEEP is shown in Figure 3. For a given stretch bending problem, the inputs contain a 3D-shaped workpiece, a 3D-shaped mold, and the motion parameters of the working arms. The output is the final shape of workpiece. Initially, we preprocess the data to structure the inputs and output, and design several encoders to extract property-aware tokens. We propose a two-stage, Transformer-based (Vaswani et al., 2017) module, the Deformation Predictor (DP), to effectively

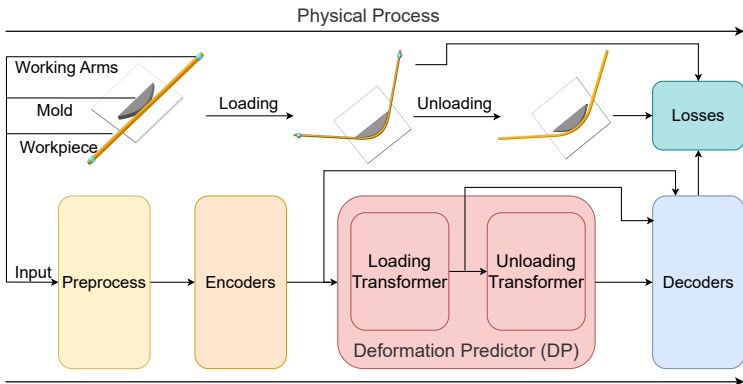

Figure 3: The overview of LaDEEP. See Appendix A.1 for detailed framwork and model structure.

characterize physical interactions between objects. Finally, we decode high-dimensional tokens back to the original space to calculate distinct losses and optimize different parts of the network.

## 2.1 PREPROCESS

The formats of the considered objects in our case are unstructured and unsuitable for deep learning models. We first need to preprocess these objects to organize their structure, reduce redundancy, and ensure completeness. In practice, most workpieces are slender with a constant cross-sectional shape along their length. This means the cross-section remains the same when cut parallel to certain faces of the workpiece, allowing it to be recorded as a 2D image. We define curves along the length of the workpiece as "characteristic lines", which contain curvature information and describe how the cross-section expands. A 2D cross-section and a characteristic line can fully represent the 3D workpiece. For the 3D mold, its cross-section is designed based on the workpiece, so it can be represented simply by its characteristic line. We sample each characteristic line (for both the workpiece and the mold) as a point set $\mathbf{p} = \{p_i | i = 1, \cdots, M\}$, consisting of $M$ points where each point $p_i$ is a position vector of $(x, y, z)$. Regarding the motion of each working arm, it applies loads on the workpiece through moving and rotating, controlled by 6 degrees of freedom (DoFs) $\{u_x, u_y, u_z, r_x, r_y, r_z\}$, where $\{u_x, u_y, u_z\}$ represent spatial displacements, and $\{r_x, r_y, r_z\}$ are rotations around each axis. In practice, most of the products are symmetric and the middle of the workpiece are kept static when processing. For simplicity and without violating the physics, we only consider the half of the system.

## 2.2 ENCODER

After processing the objects, we introduce several distinct encoders to individually encode each object. These encoders are designed based on the properties of different objects and integrate those properties into the token sequences.

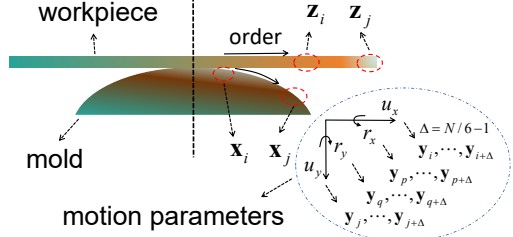

Figure 4: Property-aware tokens denoted as sequence of $\mathbf{x}$ and $\mathbf{z}$ corresponding to the regions of mold and workpiece, respectively.

**Characteristic Line Encoder (CLE)**. The characteristic line is sampled as a point set. As mentioned earlier, most workpieces are slender, and the molds follow a similar structure. Hence, the point set sampled from the characteristic line possesses an inherent property of order, which is different from point clouds and crucial for capturing different deformation behaviors. As shown in Figure 4, we encode the line into sequential region tokens, ensuring each token ($\mathbf{x}$ and $\mathbf{z}$) has explicit order implications during modeling. First, we embed each point into a high-dimensional space with an embedding size of $C$ using a linear layer. We then patchify these high-dimensional points into $N$ regions, each with $M/N$ adjacent points ($M$ is set as an integer multiple of $N$). Next, $N$ distinct linear layers are utilized to separately project these local regions into tokens $\mathbf{x}_l \in \mathbb{R}^{N \times C}$. However, these tokens capture local curvature but lack global features, such as the overall trend and length. To incorporate global features, we utilize two linear layers to embed and encode $M$ original points into a global feature $\mathbf{x}_g \in \mathbb{R}^{1 \times C}$. Then this

global feature is repeated $N$ times and added to $\mathbf{x}_l$, forming the final token sequence. We apply two separate CLEs, with non-shared parameters, to encode the characteristic lines of both the workpiece and the mold, resulting in outputs $\mathbf{x}_w, \mathbf{x}_m \in \mathbb{R}^{N \times C}$, respectively.

**Cross-Section Encoder (CSE)**. Each workpiece has a constant cross-section that is described as a 2D image with shape $1 \times H \times W$. We use the Signed Distance Function (SDF) (Guo et al., 2016) to provide a low-redundancy representation of the shape and structure for cross-section, omitting unnecessary brightness, spectrum and semantic information. Specifically, a contour set $Con$ in a domain $\Omega \subset \mathbb{R}^2$ is defined as $Con = \{(i, j) \in \mathbb{R}^2 : g(i, j) = 0\}$, where $g$ is the sign function indicating the point's position relative to the contour: $g(i, j) = -1$ when $(i, j) \in \Omega$ and $g(i, j) = 1$ when $g(i, j) \in \complement_{\mathbb{R}^2}\Omega$. The SDF $D(i, j)$ is formulated as $D(i, j) = \min_{(i', j') \in Con} |(i, j) - (i' - j')| \cdot g(i, j)$, which measures the distance of a given point $(i, j)$ to the nearest contour point with a sign indicting the relative position. After getting the SDF representation $\mathbf{s} \in \mathbb{R}^{1 \times H \times W}$, a frozen pre-trained ResNet (He et al., 2016) is utilized as a backbone to extract features, commonly applied in Computer Vision (CV) (Voulodimos et al., 2018). Since the backbone is trained on natural images and may not be suitable for the SDF, we add several convolutional layers with trainable parameters for greater flexibility. The cross-section of the workpiece is taken as input, resulting in the flattened output feature $\mathbf{s}_w \in \mathbb{R}^{1 \times C}$.

**Object Feature Fusioner (OFF)**. Recall that a workpiece is described by two separated features, the cross-section $\mathbf{s}_w \in \mathbb{R}^{1 \times C}$ and the characteristic line $\mathbf{x}_w \in \mathbb{R}^{N \times C}$. We then fuse them to obtain complete workpiece representation. We repeat the feature $\mathbf{s}_w$ by $N$ times and add it into $\mathbf{x}_w$ to form the fused result $\mathbf{z}_w \in \mathbb{R}^{N \times C}$. In this way, on the one hand, we can ensure that each token in $\mathbf{x}_w$ contains cross-sectional information, preserving the knowledge of constant cross-section. On the other hand, it does not disrupt the inherent order property within the feature of characteristic line $\mathbf{x}_w$.

**Motion Parameter Encoder (MPE)**. As claimed before, the loads applied on the workpiece are caused by the movement of each working arm, which is controlled by 6 DoFs. We use multiple tokens to capture the comprehensive influence of the movement effect along each degree of freedom. Specifically, the DoF of each axis is embedded with size $C$ and then separately projected into $N = Y \times 6$ tokens in total by 6 distinct linear layers ($N$ is set as an integer multiple of 6). Each DoF is represented by adjacent $Y$ tokens, distributing the effects across different tokens for better learning. The final output are the tokens $\mathbf{y}_m \in \mathbb{R}^{N \times C}$, representing the motion parameters.

We finally obtain three property-aware sequences corresponding to the mold $\mathbf{x}_m$, the motion parameters $\mathbf{y}_w$ and the workpiece $\mathbf{z}_w$. They share the same shape with $N \times C$, where $N$ is the number of tokens and $C$ is the embedding size. Without introducing ambiguity and for simplicity, we remove the subscript notation and use $\mathbf{x}, \mathbf{y}, \mathbf{z}$ to represent these 3 sequences, respectively. Figure 4 illustrates the stretch-bending process on the $xoy$ plane and indicates the corresponding tokens. Each object's modeling retains its distinct physical structure, and each token contains specific inherent properties.

## 2.3 Deformation Predictor

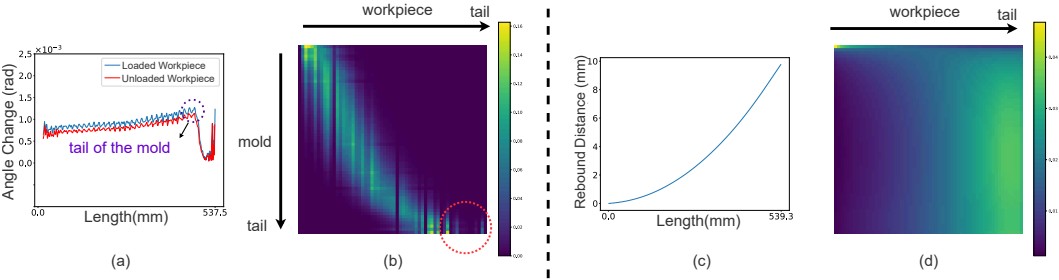

(a)                    (b)                   (c)                   (d)

Figure 5: Visualization of attention maps in DP. (a) The angle change of along the length of the workpiece. (b) The contact points will be assigned higher weights. The red circle refers to the part of the workpiece that exceeds the mold. (c) The rebound distance along the length of the workpiece. (d) The weights change smoothly. Higher weights are assigned to parts with more rebound distance.

In traditional FEM method, the process is treated locally with incremental approximations. However, with the powerful nonlinear modeling capabilities of deep learning, we can approximate solutions globally, accounting for interaction across the entire structure. As we have encoded the sequentially partitioned objects into property-aware token sequences, which contain essential order property of

corresponding solids, it is natural to leverage the Transormer (Vaswani et al., 2017), which excels at capturing global relationships and long-range dependencies, to explore the global interactions with the order property. Hence, We propose the Deformation Predictor (DP) – a two-stage, Transformer-based module, to effectively capture complex interactions between objects and approximate solutions with property-aware token sequences.

In the first stage, we employ Transformer with cross-attention mechanism (Chen et al., 2021) to model the relationships between objects in the loading stage. During this process, the working arm moves and rotates, exerting loads on the workpiece, causing it to lengthen and curve over a mold. Both the motion parameters and the mold affect the deformation of the workpiece through movement along 6 DoFs and complex nonlinear, non-smooth contact. Recall that $\mathbf{x}$, $\mathbf{y}$ and $\mathbf{z}$ respectively represent the property-aware token sequences of the motion parameters, the mold and the workpiece, the relationships are modeled in the $i$-th layer as:

$$(\mathbf{x}_0, \mathbf{y}_0, \mathbf{z}_0) = (\mathbf{x} + \mathbf{x}_{pos}, \mathbf{y} + \mathbf{y}_{pos}, \mathbf{z} + \mathbf{z}_{pos})$$
$$(\mathbf{q}_{i-1}, \mathbf{k}_{i-1}, \mathbf{v}_{i-1}) = (\mathrm{Linear}(\mathrm{concat}(\mathbf{x}_0, \mathbf{y}_0)), \mathrm{Linear}(\mathbf{z}_{i-1}), \mathrm{Linear}(\mathbf{z}_{i-1})) \tag{1}$$
$$\mathbf{z}'_{i-1} = \mathrm{LN}(\mathbf{z}_{i-1} + \mathrm{softmax}(\frac{\mathbf{q}_{i-1}\mathbf{k}_{i-1}^T}{\sqrt{C}})\mathbf{v}_{i-1}), \ \mathbf{z}_i = \mathrm{LN}(\mathbf{z}'_{i-1} + \mathrm{FFN}(\mathbf{z}'_{i-1}))$$

Here, $\mathbf{x}_{pos}, \mathbf{y}_{pos}, \mathbf{z}_{pos} \in \mathbb{R}^{N \times C}$ are position embeddings. Linear$(\cdot)$ is a linear projection layer, and $\mathbf{q}, \mathbf{k}, \mathbf{v} \in \mathbb{R}^{N \times C}$. The $j$-th row of $\mathbf{q}$ contains information of the $j$-th region in the mold and the $\lfloor j/Y \rfloor$-th DoF of the motion parameters. The $j$-th rows of $\mathbf{k}$ and $\mathbf{v}$ represent the $j$-th region in the workpiece. Consequently, through the cross-attention, the element of the $j$-th row and the $k$-th column of the attention weight matrix represents the influence of the combined action of the $\lfloor j/Y \rfloor$-th DoF's motion and the $j$-th region of the mold on the $k$-th region of the workpiece. And thus, *the attention mechanism properly characterizes physical interaction between the mold and workpiece, which rationalizes the usage of Transformer architecture*. There are $S^a$ layers in total. Throughout the process, both the mold and the motion parameters remain invariant. Therefore, we use the concat$(\cdot)$ of the initial $\mathbf{x}_0$ and $\mathbf{y}_0$ as inputs for each layer, only forwarding $\mathbf{z}_0$. The final output, $\mathbf{z}^a \in \mathbb{R}^{N \times C}$, is the high-dimensional representation of the workpiece after loading.

In the second stage, we use Transformer with self-attention mechanism (Vaswani et al., 2017) to learn the rebound of the workpiece. During this process, the working arm is released, allowing the elastic part of the deformation in the workpiece to recover, causing a rebound. Since only the workpiece itself is involved, we utilize self-attention to model this stage as follows:

$$\mathbf{z}_0 = \mathbf{z}^a + \mathbf{z}^a_{pos}, \ (\mathbf{q}_{i-1}, \mathbf{k}_{i-1}, \mathbf{v}_{i-1}) = (\mathrm{Linear}(\mathbf{z}_{i-1}), \mathrm{Linear}(\mathbf{z}_{i-1})\mathrm{Linear}(\mathbf{z}_{i-1}))$$
$$\mathbf{z}'_{i-1} = \mathrm{LN}(\mathbf{z}_{i-1} + \mathrm{softmax}(\frac{\mathbf{q}_{i-1}\mathbf{k}_{i-1}^T}{\sqrt{C}})\mathbf{v}_{i-1}), \ \mathbf{z}_i = \mathrm{LN}(\mathbf{z}'_{i-1} + \mathrm{FFN}(\mathbf{z}'_{i-1})) \tag{2}$$

This is the forward in the $i$-th layer. We project $\mathbf{z}_0$ into $\mathbf{q}, \mathbf{k}, \mathbf{v} \in \mathbb{R}^{N \times C}$ as the inputs of self-attention. The $j$-th rows of $\mathbf{q}$, $\mathbf{k}$ and $\mathbf{v}$ contain information of the $j$-th region in the workpiece. Through self-attention, the element of the $j$-th row and the $k$-th column of the attention weight matrix represents the influence of the $j$-th region on the $k$-th region of the workpiece in the rebound stage. There are $S^b$ layers in total, and the final output, $\mathbf{z}^b \in \mathbb{R}^{N \times C}$, is the high-dimensional representation of the workpiece after unloading.

Toward an intuitive understanding of DP, we visualize the attention maps. There exists a noticeable corner (Figure 5a) corresponding to the tail end of the mold, where the workpiece extends beyond the mold, resulting in a distinct abrupt change. As shown in Figure 5b, the loading module learns the interaction pattern of the mold on the workpiece due to the explicit inherent order property in the input token sequences: the interaction is most evident at the contact points between the mold and the workpiece, which are assigned higher weights, while the part of the workpiece extending beyond the mold is minimally affected. Figure 5c and 5d show that the closer it is to the tail of the workpiece, the greater the rebound. This is consistent with the pattern learned by the unloading module: the attention weights change smoothly, with greater weights assigned as it is closer to the tail end.

## 2.4 DECODER AND LOSS FUNCTION

After getting the high-dimensional representations of the workpiece, we develop several decoders to map high-dimensional representations back to the original space and calculate losses.

**Characteristic Line Decoder (CLD)**. We roughly reverse the structure of the Characteristic Line Encoder (CLE) as Characteristic Line Decoder (CLD) to decode the high-dimensional representation back to the original space. Take $\mathbf{z}^b \in \mathbb{R}^{N \times C}$ from unloading module as input, we first use $N$ distinct linear layers to decode each token to adjacent $M/N$ points. Then, a linear layer is utilized to de-embed these points back to the Euclidean space. We decode $\mathbf{z}^a \in \mathbb{R}^{N \times C}$ from loading module in the same way. The final outputs for $\mathbf{z}^a$ and $\mathbf{z}^b$ are $\mathbf{p}^a \in \mathbb{R}^{M \times 3}$ and $\mathbf{p}^b \in \mathbb{R}^{M \times 3}$, respectively.

**Cross-Section Decoder (CSD)**. Given that the cross-section has a significant impact on the forming of the workpiece (Yu et al., 2018), we establish a Cross-Section Decoder (CSD) to recover SDFs, ensuring the extracted features by Cross-Section Encoder (CSE) are effective. We refer to the structure of VAE (Kingma & Welling, 2022) and utilize deconvolution layers and interpolation operations to construct CSD. Take $\mathbf{s}^w \in \mathbb{R}^{1 \times C}$ from CSE as input, the output is $\mathbf{s}_r \in \mathbb{R}^{1 \times H \times W}$, matching the shape of ground truth $\mathbf{s} \in \mathbb{R}^{1 \times H \times W}$.

**Loss Functions** Our approach utilizes two distinct loss functions: the reconstruction loss $loss_r$, for the cross-section, and the prediction loss $loss_p$, for the workpiece. The $loss_p$ is calculated twice, once for the workpiece after loading and once after unloading. We use 3 optimizers to train different parts of the network with corresponding losses. For the reconstruction loss, we measure the Mean Square Error (MSE) $loss_r = \frac{1}{n} \sum_1^n (\mathbf{s} - \mathbf{s}_r)^2$. The function $loss_p$ measures the discrepancy between the characteristic lines. Due to imbalanced value distributions across different coordinate axes of the 3D characteristic line, normalizing all axes equally could lead to incorrect shifts. To address this, we emphasize axes with more significant orders of magnitude by employing a coordinated L2 loss defined as: $loss_p = \lambda_x \|\Delta x\|_2 + \lambda_y \|\Delta y\|_2 + \lambda_z \|\Delta z\|_2$, where $\lambda_x$, $\lambda_y$ and $\lambda_z$ are weights corresponding to 3D axes, computed based on the data range along each axis. The terms $\Delta x$, $\Delta y$ and $\Delta z$ denote the differences between the ground truth $\tilde{\mathbf{p}}^a$ and $\tilde{\mathbf{p}}^b$ and the prediction $\mathbf{p}^a$ and $\mathbf{p}^b$ along each respective axis.

## 3 EXPERIMENTS

### 3.1 EXPERIMENT SETTINGS

**Dataset**. We employ FEM (Reddy, 2019) with fine mesh to generate highly accurate dataset. This dataset contains 3000 samples and each sample contains: a characteristic

Figure 6: Five types of cross-section, which are indexed 1-5 from left to right.

and a cross-section of the 3D workpiece, a characteristic of the 3D mold, and the motion parameters. For the cross-section, we select 5 representative types of cross-section structures, depicted in Figure 6, which can cover most of the practical products. There are random parameters that control the arc radius, arc radian, thickness and height. For the characteristic line of the workpiece, we place the initial straight workpiece on the x-axis from original point to the maximal length and randomly sample the length. Regarding the mold, we utilize two 1/4 elliptical arcs on two perpendicular planes to combine a 3D curve as the characteristic line. The motion parameters are calculated by classical involute approach (Arnold et al., 2012) with the characteristic line of the mold. We use Abaqus (Khennane, 2013) software to perform the computation. The dataset is split into an 8:1:1 ratio for training, evaluation and test. More details are in Appendix A.3.

**Baselines.** The compared baselines are classified into two groups, the traditional numerical methods, and the deep learning methods. For traditional methods, we select FEM as the representative, and for existing DL methods, we comprehensively compare LaDEEP with 10 well-known models: DeepONet (Lu et al., 2019), FNO (Li et al., 2022), GINO (Li et al., 2024b), SFNO (Bonev et al., 2023), TFNO (Kossaifi et al., 2023), UNO (Rahman et al., 2022), FactFormer (Li et al., 2024a), LSM (Wu et al., 2023), Transolver (Wu et al., 2024) and TCN (Oord et al., 2016) which are well-known models of PDE solvers. See Appendix A.4 for more comprehensive description.

**Metrics**. We utilize three metrics to evaluate LaDEEP from different aspects: MAD (Mean Absolute Distance), IoU 3D (Intersection over Union) and TE (Tail Error) for evaluation. Details about these metrics are described in Appendix A.5.

### 3.2 MAIN RESULTS

**Compared to FEM** Figure 2b shows the results of computing time and MAD between LaDEEP and traditional FEM. The FEM methods are set with various granularities of meshes (coarser than

that for dataset generation) for different computation efficiency and accuracy. For FEM, a reduced number of elements leads to improved computational speed, but at the expense of accuracy. For workpieces of the type-1 cross-section, FEM methods need around 342s for coarse-grained meshes, with errors on $2.0092mm$(MAD) and around 598s for finer-grained meshes, with $0.0568mm$(MAD). For LaDEEP, only $0.0453s$ is needed for computations with $0.1823mm$(MAD), providing around 9603 times acceleration. With the similar computation accuracy, the accelerations range from 9603 times to around 60068. In our dataset, the type-1 is the simplest, and type-4 is the most complicated one that has two arcs with different radius and arc length. The FEM needs much more computation time for type-4. But for LaDEEP, the shapes of cross-sections are all represented by SDFs with shape $512 \times 256$. The inference speed is consistent regardless of shape complexity, as computation time does not increase for more intricate profiles. We therefore observe a higher acceleration ratio.

**Compared to deep learning alternatives**   To evaluate LaDEEP from a comprehensive view, we experiment in two settings: (1) We compare LaDEEP with other naive baselines. These baselines are added with simple encoders and decoders constructed with linear layers. (2) We compare LaDEEP with other modified baselines. The encoders and decoders in LaDEEP are kept unchanged. We only replace the Deformation Predictor (DP) with these baselines. From these two settings, we can explore the effectiveness of the encoders, decoders and DP in LaDEEP. See Appendix A.4 for comprehensive description of implementation.

Table 1: Performance comparison with baselines. Columns 2-4 are results in setting (1), and columns 5-7 are results in setting (2). For MAD and TE, a smaller value indicates better performance, whereas for IoU 3D, the opposite is true. The best result is in bold and the second best is underlined. Improvement in the last column refers to the average relative error reduction across all metrics of corresponding models. Improvement in the last row refers to the relative error reduction w.r.t the second best model.

| Model | MAD(mm) | IoU 3D(%) | TE(mm) | MAD(mm) | IoU 3D(%) | TE(mm) | Improvement |
|---|---|---|---|---|---|---|---|
| DeepONet | 0.3836 | 74.43 | 0.7806 | 0.2445 | 82.52 | 0.6238 | 22.41% |
| FNO | 0.3251 | 78.15 | 0.7488 | 0.2325 | 82.27 | 0.6060 | 16.44% |
| GINO | 0.3394 | 76.65 | 0.8067 | 0.2567 | 81.08 | 0.6812 | 15.24% |
| SFNO | 0.3366 | 77.79 | 0.7737 | 0.2362 | 82.26 | 0.6079 | 19.07% |
| TFNO | 0.3267 | 78.03 | 0.7598 | 0.2389 | 81.59 | 0.6419 | 15.65% |
| UNO | 0.3380 | 77.56 | 0.7795 | 0.2137 | 83.33 | 0.5810 | 23.23% |
| FactFormer | 0.3458 | 78.28 | 0.8134 | 0.2404 | 82.17 | 0.6478 | 18.60% |
| LSM | 0.3356 | 77.86 | 0.7222 | 0.2099 | 84.00 | 0.4987 | 25.43% |
| Transolver | 0.3912 | 76.31 | 0.8359 | 0.2052 | 84.21 | 0.5422 | 31.03% |
| TCN | 0.4047 | 73.12 | 0.8080 | 0.2585 | 78.61 | 0.6431 | 21.35% |
| LaDEEP | **0.1698** | **86.58** | **0.4591** | **0.1698** | **86.58** | **0.4591** | / |
| Improvement | 47.77% | 10.64% | 36.43% | 17.25% | 2.81% | 7.94% | / |

As presented in Table 1, LaDEEP performs consistent state-of-the-art in both settings. Notably, from setting (1) to (2), these baselines also gain significant improvement (20.85% on average) with proposed encoders and decoders, demonstrating the effectiveness of our design in handling large deformations of elastic-plastic solids. Also, some advanced Transformer-based models, such as LSM and Transolver, achieve impressive improvement (25.43% and 31.03%) after taking property-aware token sequences as inputs. This is because these models are designed for general cases, without considering problem-specific properties, such as the order of the sequentially partitioned

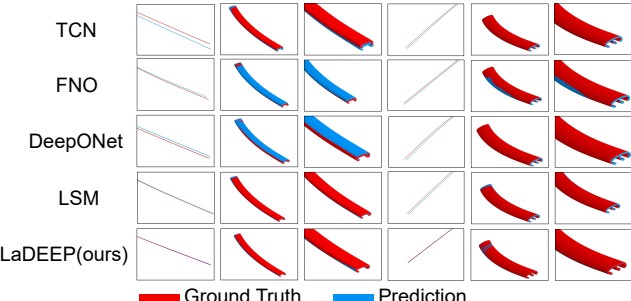

Figure 7: Visualization of results in setting (2). Column 1 and 4 are characteristic lines. Column 2-3 and column 5-6 are workpieces corresponding to column 1 and 4, respectively. We only select part of results to visualize due to the space limitation. More visualizations are in Appendix A.9.

regions on the slender workpiece. After incorporating the property-aware token sequences, these models are more effective to characterize the interactions between objects. Additionally, they perform better then neural operators due to the strong capacity of Transformer in global modeling and long-distance dependency. However, the considered case is involved two stage – loading and unloading. Without staged modeling approach, they struggle to approximate the solutions accurately. Neural operators also face the same dilemma, although they also raise some improvements with our designed modules. As shown in Figure 7, the deviations in the predictions are primarily due to discrepancies in length (row 2, column 1-3) and shifts in curve positions (row 2, column 4-6). Benefit from analyzing the inherent properties of the case, LaDEEP can effectively capture underlying correlation between objects. We thereby achieve superior performance among all models. The analysis of samples with large errors is in Appendix A.6.

Table 2: Results of Cross-section Generalization.

| Setting ID | MAD(mm) | IoU 3D(%) | TE(mm) |
|---|---|---|---|
| 1 | 0.4122 | 72.02 | 1.1084 |
| 2 | 0.2232 | 84.56 | 0.6153 |
| 3 | 0.2313 | 83.78 | 0.6341 |

**Transferability** We conduct experiments to explore the transferability of LaDEEP to unseen cross-sections. We keep the training parameters the same for all experiments except the training epoch. The training setting are: (1) **Zero-shot Testing**: We use the first 4 out of 5 kinds of cross-section as training data and the 5th unseen cross-section as test data. The model is trained for 600 epochs.(2) **Fine-tuning**: We further investigate by splitting the 5th cross-section into an 8:2 ratio for training and test. We fine-tune the model in (1) for additional 200 epochs. (3) **Full Dataset Training**: We evaluate the model on the 5th cross-section using the model trained with all types of cross-sections. The results are shown in Table 2 which demonstrate that while the LaDEEP exhibits some zero-shot capabilities, there is room for improvement. Fine-tuning significantly enhances performance, achieving results close to the baseline model trained with all cross-sections. This highlights LaDEEP's potential in practical applications. When new data with unseen cross-section appears, fine-tuning the basic model to generalize the data is a considerable way. See Appendix A.7 for another experiment about the transferability on new materials.

### 3.3 ABLATION STUDIES

The proposed model comprises several essential components and we assess their efficacy through comprehensive ablation studies. We consider four distinct variants including encoders, fusioner, predictor and loss function. The settings are: 1) Replace the Object Feature Fusion (OFF) with attention mechanism which is also a global fusioner; 2) Replace the SDF with gray image. 3) Replace the CLE with PointNet (Qi et al., 2017) containing max operation that omits the order information 4) Replace the 1st stage of the DP with MLP (Pinkus, 1999) to determine whether the global modeling and long-distance dependency are important; 5) Replace the 2nd stage of the DP with MLP; 6) Replace the $loss_p$ with Mean Square Error (MSE) which gives equal equations to all axes.

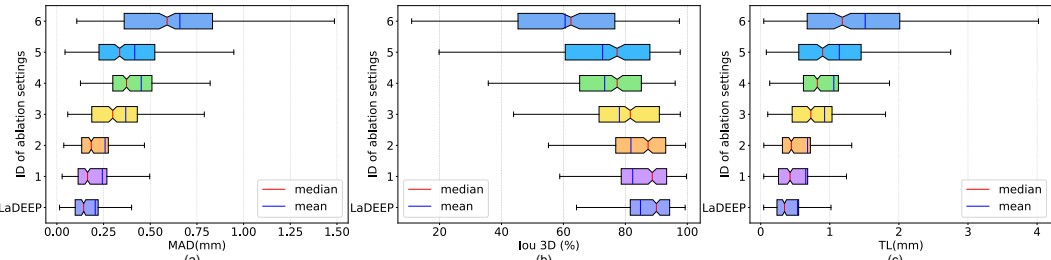

Figure 8: Results of ablation studies. The indexes on the y-axis are corresponding to the ablation setting indexes. For MAD and TE, a smaller value indicates better performance, whereas for IoU 3D, the opposite is true.

As shown in Figure 8, our proposed essential modules synergistically boost the modeling capacity and performance. Compared with attention-based fusioner, while the accuracy is close, the computation times are $67.9ms$ (LaDEEP) and $86.1ms$ (attention-based) with batch size 8, respectively. The extra $26.8\%$ runtime penalty does not pay off. Using gray images of cross-sections as input lowers down the accuracy by $25.77\%$ on MAD caused by the redundant information of gray images. The consideration about the order of the points sampled from the characteristic line is significant. The

expanded overall error distribution in the results indicates a shift in the implicit physics learned by the model. We can observe the same phenomenon if we replace two modules in DP with MLP, respectively. Due to the lack of the explicit physical modeling, the inductive bias of the model has shifted. This shift can lead to the model capturing incorrect dependencies and not learning accurate physical knowledge. The results of MSE loss are much worse. Note that the characteristic line is 3D. The coordinate components on 3 axes are imbalanced. The results of MSE loss proves that the coordinated L2 loss can effectively alleviate this problem.

### 3.4 DEPLOYMENT

We deploy LaDEEP into a real manufacturing factory with aluminum stretch bending fabrications. The FEM method has been used for computations over years but provides limited value due to the bad computation efficiency – a practical workpiece requires days for a single FEM computation and the whole design process (iterative simulation) holds for 2-3 months. This gap motives us to design more efficient, deep learning-based surrogate models.

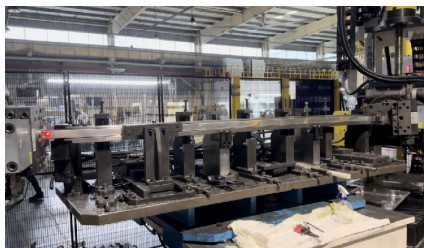

Figure 9: A mold designed based on LaDEEP is used for on-site production.

In real deployments, there are always inevitable errors between the computations and productions due to the inherent complexities and uncertainties in the manufacturing process. These errors vary from different product scenarios with many sources such as the mold production error, the mold installation error, the motion parameters error, the material property variation error and etc. These errors can be additive or canceling, resulting an overall application error (AE) around $1mm \sim 10mm$ (e.g., a workpiece with length 2m, the relative error is around 0.5%). Then, such application error will be compensated by empirical production techniques and feedback to computations, ensuring the final production error (PE) within $1mm$. The empirical production techniques include machining of the mold by tools directly, as well as fine-tuning the parameters of working arms.

We deploy LaDEEP into practical scenarios and develop a two-cycled mold design paradigm as shown in Figure 18 based on LaDEEP. More details are described in Appendix A.8. We accomplish seven real-product designs with LaDEEP and the average application error is around 8.5mm. This is sufficient for on-site adjustment. Tow of

Table 3: PE of two products designed by our two-loop mold design paradigm.

| Product | DX11-RQT | H93-FUQT |
|---------|----------|----------|
| MAD(mm) | 0.32 | 0.29 |

them are corrected by on-site adjustment and the final production errors are shown in Table 3. The mold design process is reduced to around 1 week and the efficiency is improved by around 8-10 times. Figure 9 is a mold designed based on LaDEEP used for on-site production.

## 4 CONCLUSION AND DISCUSSION

In this paper, we firstly attempt to apply deep learning into large deformations of elastic-plastic solids. We propose LaDEEP, a novel deep learning-based framework tailored for an industrial task, stretch bending. We design several modules to encode sequential property-aware tokens and propose a two-stage, Transformer-based module, the Deformation Predictor (DP) to approximate the two-stage solutions. We generate a dataset to support our approach and fill the data gap in this area. LaDEEP achieves five magnitudes faster speed than FEM with a comparable accuracy, and gains 20.47% relative improvement on average compared to other deep learning baselines.

**Limitation.** We are well aware that there exists a limitation of the application scenarios of LaDEEP. However, due to the general insights of exist models, directly applying them into the reality-related complex large deformations of the elastic-plastic solids without problem-specific property would be hard to capture the correlations among solids accurately. Hence, we take the industrial technique, stretch bending, as a good start point to explore the problem properties and incorporate them into the models to learn effective inductive biases. More complex problems and general solvers would be studied in future works. We believe that the framework of LaDEEP is a meaningful modeling approach for solids involving large elastic-plastic deformations and contacts, and has great potential of being adapted to other complex industrial applications.

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

# A APPENDIX

## A.1 LaDEEP FRAMEWORK

The detailed overview of LaDEEP framework is illustrated in Figure 10. The whole framework contains three main components.

**Data Preprocess**. In this process, we need to process the data to represent the assembly model without redundant information. For a given stretch-bending problem, the inputs include a 3D-shaped workpiece, a 3D-shaped mold, and the motion parameters of the working arms. In practice, most of the products are symmetric. For simplicity and without violating the physics, we only consider the simulating the half of the system and keep the middle of the workpiece static. We use the combination of the cross-section and the characteristic line to represent the workpiece. The mold is only represented by the characteristic because its cross-section is related to the cross-section of the workpiece. The movement of the working arms is depicted by a vector with size of 6, representing 6 degree-of-freedom motion.

**Deep Learning-based Surrogate Model**. The detailed structure of the model is depicted in Figure 11.Take the processed representations as inputs, the model use encoders (CLE, CSE, MPE) to extract the high-dimensional features. Then a fusioner (OFF) is utilized to fuse the two features that represent the cross-section and characteristic line of the workpiece, respectively. The feature of the cross-section is reconstructed by a decoder(CSD). Then the result is used to calculate the loss function $loss_r$ and optimize the part of the model (CSE and CSD). Subsequently, the Deformation Predictor (DP) is composed of two-stage modules, loading and unloading. The outputs of the loading module and unloading module are projected back to the original space through decoders. Then the outputs are used to calculate the losses $loss_p$ to train corresponding modules. See Appendix A.2 for detailed training strategy.

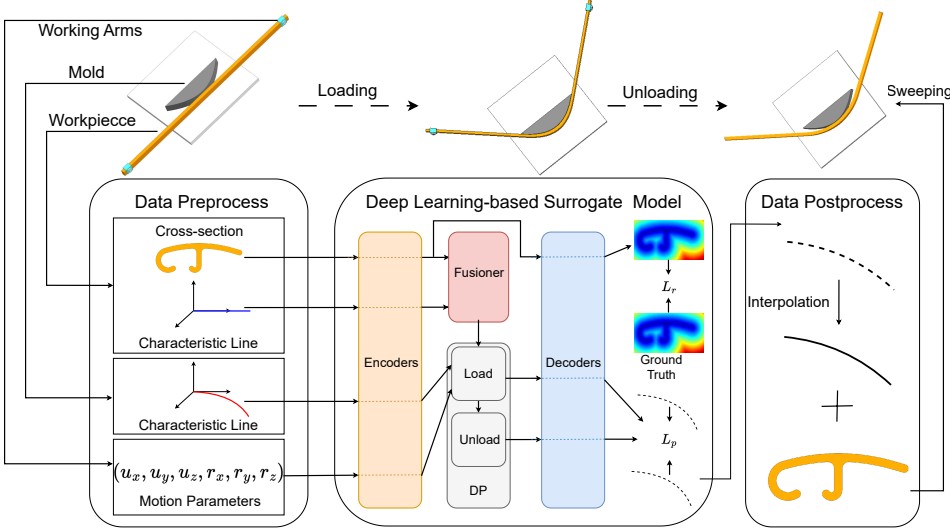

Figure 10: The detailed overview of LaDEEP framework.

**Data Postprocess**. After getting the deformed and rebounded characteristic line of the workpiece, we reconstruct the original three-dimensional workpiece for practical usage. We first use the interpolation operation to transform the sampled characteristic line into continuous form. Then with the two-dimensional cross-section of the workpiece, we use sweeping operation to reconstruct the original workpiece. The basic notion embodied in sweeping operation is that a set moving through space may trace or sweep out volume (a solid) that may be represented by the moving set and its trajectory. More specifically, we move the cross-section along the direction of the characteristic line from the front to the end. In the sweep process, the cross-section is perpendicular to the tangent of the characteristic line at the arrived point.

## A.2 MODEL CONFIGURATION AND TRAINING DETAILS

The detailed structure of LaDEEP model is depicted in Figure 11. For simplicity and clarity, we set batch size as 15 and don't emphasize it in the subsequent description. For the input features, the size of SDF is $512 \times 256$ with a single channel, the differential of characteristic lines for workpiece and mold are the same shape $300 \times 3$ and the motion parameters are a 1D vector with length 6. After being encoded by the corresponding modules, all high-dimensional features have the shape of $60 \times 64$. For modules in Deformation Predictor (DP), both of the layer number for loading module and unloading module are 3. The number of heads is 4 and the MLP expension ration is 2. The latent state for the feature from DP is the same shape as the input features as $60 \times 64$. Then after decoded, the final output has the shape of $300 \times 3$.

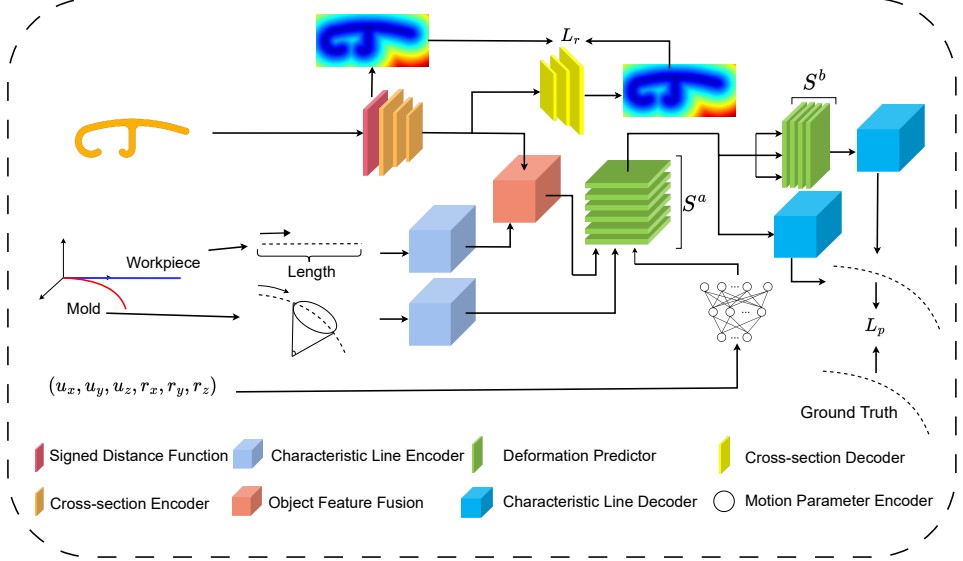

Figure 11: The detailed structure of LaDEEP.

When training LaDEEP, we employ there separate optimizers. One of them is to optimize the CSE and CSD through the loss function $loss_r$. For the other two optimizes, one is to optimize the unloading part (Unloading module and corresponding CLD), while the other focuses on optimizing the loading part and the rest (CLE, MPE, OFF, Unloading module and corresponding CLD). They are all Adam and we set learning rate 1e-3, weight decay 5e-5 the same for them. We use cosine annealing schedule to update the learning rate with Cosine half-cycle decay and the minimal learning rate 1e-6. The batch size for training is 15 and the epoch is 1000. We use a sigle RTX 3090 GPU to train LaDEEP and other deep learning baselines. The training time of LaDEEP with above hyperparameters is 3.5 hours on average. More details can be found in our implemented code. The training and evaluation losses in a training process are shown in Figure 12.

## A.3 DATASET GENERATION

The training of deep learning models requires sufficient amount of data, which can hardly be obtained from real manufacturing environment. We employ traditional FEM method with fine mesh resolution to generate highly accurate dataset. Each sample is composed of three components, the 3D workpiece, the 3D mold, and the motion parameters. We collect both shapes of the workpieces after loading and unloading.

- **3D Workpiece**. The workpieces are determined by the cross-sections and characteristic lines. The cross-sections will affect the overall structural force distribution. For practical concerns, we select 5 representative types of cross-section structures, as depicted in Figure 13, from practical observation. These cross-sections can cover most practical products in term of the topology structure and size. Each kind of cross-section has different number, radian and radius of arcs, various thicknesses and heights. In each kind of cross-section, the radius of

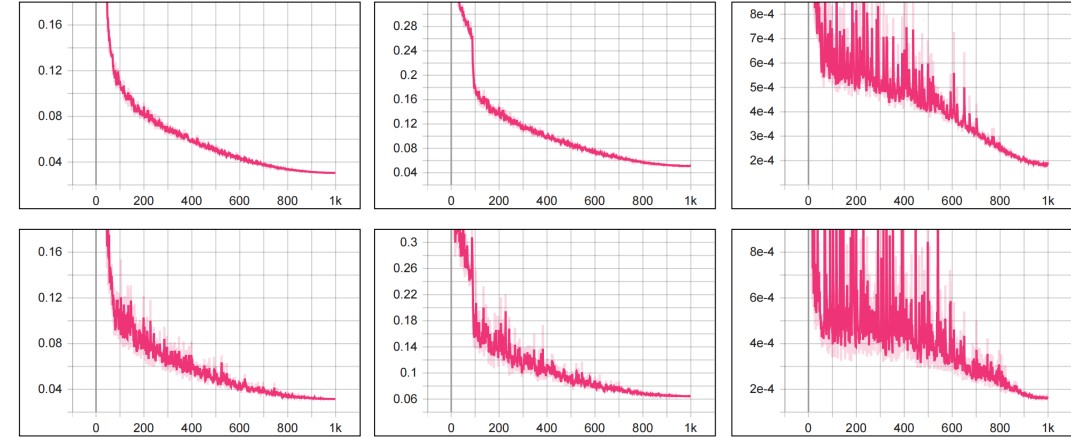

Figure 12: The training and evaluation loss curves of LaDEEP. For all sub-figures, the x-axis is the epochs and the y-axis is the corresponding loss. From top to bottom, from left to right, they are: the training $loss_p$ of loading part, the training $loss_p$ of unloading part, the training $loss_r$, the evaluation $loss_p$ of loading part, the evaluation $loss_p$ of unloading part and the evaluation $loss_r$. The smooth parameter is 0.5.

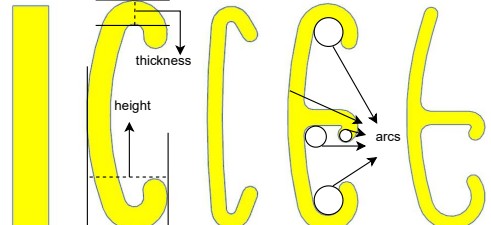

Figure 13: Five representative types of cross-section from practice. They are indexed 1-5 from left to right. Each kind of cross-section has different number, radian and radius of arcs, various thicknesses and heights.

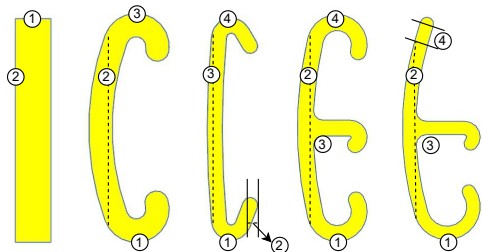

Figure 14: The indexes on each kind of cross-section are marks indicating that some parameters of the corresponding parts are sampled from specific distributions. The specific distributions are listd in Table 4.

arcs, the height and thickness are also sampled from specific distributions. The concrete configurations are listed in Table 14. For each type of cross-section structure, we generate 600 different samples. For the characteristic line of the workpiece, we place the initial straight workpiece on the x-axis from original point to the maximal length and randomly sample the length from a uniform distribution $\mathcal{U}[505, 550]$ (unit: mm).

- **3D Mold**. In order to ensure the workpiece can contact the mold tightly during the deformation, the characteristic lines for the mold should be smooth and convex. We generate two 1/4 elliptical arcs separately on two perpendicular 2D planes, then combine them into a 3D curve in space. These different elliptical arcs are determined by different ellipse parameters with uniform distributions for varying curvatures. Consider a 2D elliptic formula $\frac{x^2}{a^2} + \frac{y^2}{b^2} = 1 (a > b > 0)$ and let $(c, 0)$ be its focus point, we use three uniform distributions to control the generation, which are: $\frac{c}{a} \sim \mathcal{U}[0.1, 0.3]$, $\frac{b}{a} \sim \mathcal{U}[0.1, 0.3]$ and $a \sim \mathcal{U}[700, 900]$ (unit: mm).

- **Motion parameters**. The motion parameters consist of 6 degrees of freedom. including spatial displacement $(u_x, u_y, u_z)$, and the rotations $(r_x, r_y, r_z)$. They are calculated by a classical involute approach (Arnold et al., 2012) based on the characteristic line of the mold.

With all the 3000 sets of data, Abaqus (Khennane, 2013; Abaqus, 2011), a software based on FEM, is applied to perform the computations. The mold is set to be rigid, and the workpiece is set as elastic-plastic aluminum. In Abaqus setup, the explicit dynamics solver (Barbero, 2023) is adopted

for simulation calculation and the implicit static solver (Barbero, 2023) is for the rebound calculations. Appendix A.4 describes more settings about the FEM model.

Table 4: Distributions that control the parameters for each kind of cross-section. The indexes in the table are corresponding to those in Figure 14. The whole cross-section can be calculated through given parameters. Let the positive direction of the x-axis be $0°$ and the clockwise is the positive direction. The postfix "s" means start arc and "e" means end arc.

| Type ID | Radius Distribution (mm) | Radian Distribution (°) | Length Distribution (mm) |
|---|---|---|---|
| 1 | / | / | thickness: $\mathcal{U}[2.5, 4]$
1: $\mathcal{U}[2.5, 4]$
2: $\mathcal{U}[17, 20]$ |
| 2 | 1: $\mathcal{U}[2.5, 3]$ | 1s: $\mathcal{U}[95, 110]$
1e: $\mathcal{U}[280, 310]$
3s: $\mathcal{U}[240, 270]$ | thickness: $\mathcal{U}[1.8, 2.2]$
2: $\mathcal{U}[14, 16]$ |
| 3 | 1: $\mathcal{U}[1.4, 1.6]$ | 1s: $\mathcal{U}[95, 110]$
1e: $\mathcal{U}[230, 250]$
4s: $\mathcal{U}[290, 310]$ | thickness: $\mathcal{U}[0.8, 1.2]$
2: $\mathcal{U}[0.4, 0.6]$
3: $\mathcal{U}[14, 16]$ |
| 4 | 1: $\mathcal{U}[2.5, 3]$
3: $\mathcal{U}[1.5, 1.7]$ | 1s: $\mathcal{U}[95, 110]$
1e: $\mathcal{U}[280, 310]$
3s: $\mathcal{U}[210, 230]$
4s: $\mathcal{U}[240, 270]$ | thickness: $\mathcal{U}[1.2, 1.4]$
2: $\mathcal{U}[14, 16]$ |
| 5 | 1: $\mathcal{U}[2.5, 3]$
3: $\mathcal{U}[1.3, 1.5]$ | 1s: $\mathcal{U}[95, 110]$
1e: $\mathcal{U}[300, 310]$
3s: $[U][210, 230]$ | thickness: $\mathcal{U}[1, 1.2]$
2: $\mathcal{U}[14, 16]$
4: $\mathcal{U}[1, 2]$ |

## A.4 BASELINES

### A.4.1 CLASSICAL FEM MODELS

The considered problem involves complex large deformation of elastic-plastic solid, nonlinear solid contacts, three-dimensional motion, and nonlinear rebound. Due to the complexity and computational demands, developing a custom FEM solver from the ground up would be impractical and time-consuming. Therefore, we utilize Abaqus (Khennane, 2013; Abaqus, 2011), a robust and widely adopted industrial simulation software based on FEM, to generate our data. Each assembly model is divided into around 20,000 elements. And as a comparable baseline, we use scale factors {0.8, 0.6, 0.4, 0.2} to reduce the number of meshes in the highest resolution assembly models. During the loading stage, we employ the explicit dynamic algorithm (Barbero, 2023) for forward calculations based on dynamic equations. For the unloading stage, we use the implicit static algorithm (Barbero, 2023), iteratively solving the problem with Newton's method (Kelley, 2003).

Since the explicit dynamic algorithm in Abaqus does not support GPU computation, we leverage a 32-core CPU server for parallel processing using MPI (Walker & Dongarra, 1996). We also assess the impact of different core counts on computation speed and find that 32 cores nearly reach the maximum acceleration for our single assembly model. Increasing the core count further lead to diminishing returns due to higher communication overhead. We recognize the importance of ensuring a fair comparison between the FEM and Deep Learning models. Given the different nature of parallelization (CPU vs. GPU) and the inherent computational requirements of the FEM method, we take all reasonable steps within the constraints of our available tools and hardware to optimize the FEM simulations. We believe these measures demonstrate a conscientious approach to achieving a fair and balanced comparison, considering the specific capabilities and limitations of the FEM method within the industrial context of our study.

### A.4.2 Deep Learning Models

We implement all the deep learning baselines based on their official code. All the baselines are trained and tested under the same training strategy and loss functions as LaDEEP. We align the shapes of features through "add" or "concatenate" operation with the input shape of the models. It is not a common task so there is no official setting of the model configurations. We have tried different settings to improve the performance. We run all experiments for 3 times and compute the average results and we also maintain approximately the same number of parameters.

As mentioned in Sec.3.2, we conduct the experiments under 2 kinds of settings: (1) We compare LaDEEP with other naive baselines. These baselines are added with naive encoders and decoders constructed with linear layers. (2) We compare LaDEEP with other modified baselines. The encoders and decoders are kept unchanged. We only replace the Deformation Predictor (DP) with these baselines.

**First Setting**. The implementations of baselines, encoder and decoder in setting (1) are listed below (the parameters except mentioned below are default settings):

- **Encoders and Decoders**. For both CLE and CLD, we use four linear layers as the feature extractor, and then one linear layer as the embedding. A dropout with $p = 0.2$ is used to avoid over-fitting. The hidden activation is "ReLU" and the output activation is "Sigmoid". For MPE, we use two linear layers to align the feature shapes. Additionally, we use the same CSE structure as LaDEEP and throw out CSR.

- **DeepONet** (Lu et al., 2019). We set branch layers sizes as $\{64, 512, 1024, 512, 64\}$ and trunk layers sizes as $\{64, 512, 1024, 512, 64\}$. The activation is set as "tanh" and the kernel initializer is set as "Glorot Normal". The number of outputs is set as 64. The multiple output strategy is set as "split_both".

- **FNO** (Li et al., 2022). We set the input channel as 3 and output channel as 1. The modes are set as $\{20, 20\}$. The hidden channel is set as 64 and the number of layers is 4. The lifting channel and projection channel channel are both 64.

- **GINO** (Li et al., 2024b). We set both of the input channel and the output channel as 3. The GNO radius is set as 0.3. We set the number of the input GNO hidden layers and the output GNO hidden layers as $\{8, 8\}$. The GNO coordinate dimension is set as 3. Both of the types of input and output GNO transform are set as "linear". The modes are set as $\{8, 8, 8\}$. The lifting channel and hidden channel are both 16. The projection channel is 64.

- **SFNO** (Bonev et al., 2023). We set the input channel as 3 and output channel as 1. We set the modes as $\{32, 32\}$ and the hidden channel as 64. The type of factorization is "dense". The lifting channel and projection channel channel are both 256. The number of layers is 6.

- **TFNO** (Kossaifi et al., 2023). We set the input channel as 3 and output channel as 1. The modes are set as $\{20, 20\}$. The projection channel and the lifting channel are both 256. The hidden channel is set as 64 and the number of layers is 4. The rank is set as 0.5 and the type of factorization is set as "tucker".

- **UNO** (Rahman et al., 2022). We set the input channel as 3 and output channel as 1. The hidden channel is set as 64 and the domain padding is set as 0.2. The number of layser is set as 5. Correspondingly, for each layer, the UNO output channels are set as $\{32, 64, 128, 64, 32\}$, the UNO modes are set as $\{\{8, 8\}, \{16, 16\}, \{16, 16\}, \{16, 16\}, \{8, 8\}\}$, and the UNO scalings are set as $\{\{1.0, 1.0\}, \{0.5, 0.5\}, \{1.0, 1.0\}, \{1.0, 1.0\}, \{2.0, 2.0\}\}$.

- **FactFormer** (Li et al., 2024a). We set the both of the input dim the output dim as 64. The number of heads is set as 8 and the kernel multiplier is set as 4. The latent dim is 256.

- **LSM** (Wu et al., 2023). We set the input channel as 3 and output channel as 1. We set the dimension of model as 16. Both of the number of token and basis are set as 1. Both of the patch size and the padding are set as "1,1". The flag of using bilinear is set as "False".

- **Transolver** (Wu et al., 2024). We set the number of layers as 4 and the hidden dimension as 256. We use "relu" as the activation. The number of heads is 16 and the mlp ratio is 4. Both of the input and the output dimension are set as 64. The number of slices is set as 4 for numerical stability. The flag of using unified position is set as "False".

- **TCN** (Oord et al., 2016). We implement a 5-layer TCN and the numbers of each layer channel are set as $\{60, 180, 360, 360, 180, 60\}$. The kernel size is set as 4.

Table 5: Model parameter summary.

| Model | #Total Parameter | #Trainable Parameter |
|---|---|---|
| DeepONet | 13,814,952 | 2,638,440 |
| FNO | 15,224,617 | 4,048,105 |
| GINO | 14,169,041 | 2,992,529 |
| SFNO | 13,235,625 | 2,059,113 |
| TFNO | 13,460,245 | 2,283,733 |
| UNO | 15,091,049 | 3,914,537 |
| FactFormer | 13,811,816 | 2,635,304 |
| LSM | 13,794,377 | 2,617,865 |
| Transolver | 14,663,544 | 3,487,032 |
| TCN | 14,170,300 | 2,993,788 |
| LaDEEP(ours) | 14,260,750 | 3,084,238 |

The model parameter summary is shown in Table 6. The training and evaluation curves are shown in Figure 15.

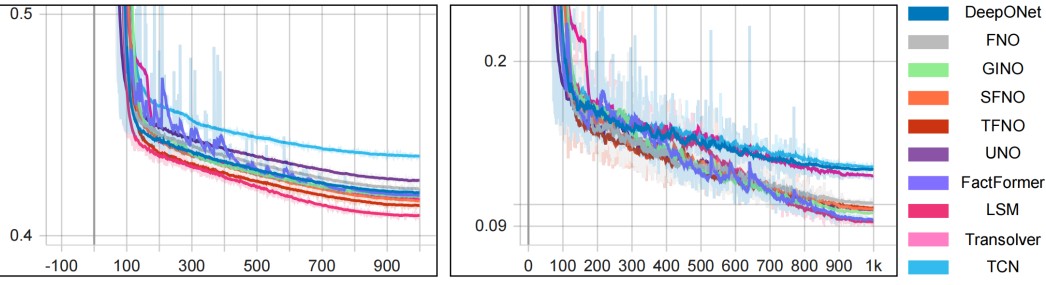

Figure 15: The training and evaluation loss curves of baselines in setting (1). For all sub-figures, the x-axis is the epochs and the y-axis is the corresponding losses. From left to right, they are: the training $loss_p$ and the evaluation $loss_p$. The smooth parameter is 0.9.

**Second Setting**. The implementations of baselines in setting (2) are listed below (the parameters except mentioned below are default settings):

- **DeepONet** (Lu et al., 2019). We set branch layers sizes as $\{64, 256, 256, 64\}$ and trunk layers sizes as $\{64, 256, 256, 64\}$. The activation is set as "tanh" and the kernel initializer is set as "Glorot Normal". The number of outputs is set as 64. The multiple output strategy is set as "split_both".

- **FNO** (Li et al., 2022). We set the input channel as 3 and output channel as 1. We use the incremental modes to control the number of parameters with modes changing from $\{2, 2\}$ to $\{16, 16\}$. The hidden channel is set as 32 and the number of layers is 4.

- **GINO** (Li et al., 2024b). We set both of the input channel and the output channel as 3. The GNO radius is set as 0.3. We set the number of the input GNO hidden layers and the output GNO hidden layers as $\{8, 8\}$. The GNO coordinate dimension is set as 3. Both of the types of input and output GNO transform are set as "linear". The modes are set as $\{8, 8, 8\}$.

- **SFNO** (Bonev et al., 2023). We set the input channel as 3 and output channel as 1. We set the modes as $\{32, 32\}$, the hidden channel as 32 and the projection channel as 64. The type of factorization is "dense".

- **TFNO** (Kossaifi et al., 2023). We set the input channel as 3 and output channel as 1. We use the incremental modes to control the number of parameters with modes changing from $\{2,$

2} to {16, 16}. The projection channel is set as 64. The hidden channel is set as 32 and the number of layers is 4. The rank is set as 0.5 and the type of factorization is set as "tucker".

- **UNO** (Rahman et al., 2022). We set the input channel as 3 and output channel as 1. The hidden channel is set as 64 and the domain padding is set as 0.2. The number of layser is set as 5. Correspondingly, for each layer, the UNO output channels are set as {32, 64, 64, 64, 32}, the UNO modes are set as {{8, 8}, {8, 8}, {8, 8}, {8, 8}, {8, 8}}, and the UNO scalings are set as {{1.0, 1.0}, {0.5, 0.5}, {1.0, 1.0}, {1.0, 1.0}, {2.0, 2.0}}.

- **FactFormer** (Li et al., 2024a). We set the both of the input dim the output dim as 64. The number of heads is set as 8 and the kernel multiplier is set as 3. The latent dim is 128.

- **LSM** (Wu et al., 2023). We set the input channel as 3 and output channel as 1. We set the dimension of model as 8. Both of the number of token and basis are set as 1. Both of the patch size and the padding are set as "1,1". The flag of using bilinear is set as "False".

- **Transolver** (Wu et al., 2024). We set the number of layers as 4 and the hidden dimension as 128. We use "relu" as the activation. The number of heads is 8 and the mlp ratio is 1. Both of the input and the output dimension are set as 64. The number of slices is set as 4 for numerical stability. The flag of using unified position is set as "False".

- **TCN** (Oord et al., 2016). We implement a 5-layer TCN and the numbers of each layer channel are set as {60, 120, 120, 120, 60}. The kernel size is set as 3.

Table 6: Model parameter summary.

| Model | #Total Parameter | #Trainable Parameter |
|---|---|---|
| DeepONet | 14,245,955 | 3,069,443 |
| FNO | 15,122,468 | 3,945,956 |
| GINO | 14,858,588 | 3,682,076 |
| SFNO | 14,329,508 | 3,152,996 |
| TFNO | 14,568,744 | 3,392,232 |
| UNO | 14,877,892 | 3,701,380 |
| FactFormer | 14,204,675 | 3,028,163 |
| LSM | 14,603,380 | 3,426,868 |
| Transolver | 14,441,971 | 3,265,459 |
| TCN | 14,333,551 | 3,157,039 |
| LaDEEP(ours) | 14,260,750 | 3,084,238 |

To ensure the comparison is apple-to-apple, we keep the encoders and decoders unchanged, only replacing the DP module with the corresponding baseline. The model parameter summary is shown in Table 6. The training and evaluation curves are shown in Figure 16.

## A.5 EVALUATION METRICS

We utilize three different metrics to evaluate the performance of the model from multiple aspects. The definition of these metrics are listed below:

- **MAD**. The mean absolute distance (MAD) measures the characteristic line distance between the prediction and ground truth of the workpiece.

- **IoU 3D**. The intersection over union (IoU) of the reconstructed workpiece compared to the ground truth objects. We utilize it to assess the real prediction accuracy at each point.

- **Tail Error (TE)**. For large deformation, the errors on the tail faces are more likely to accumulate. We therefore employ Tail Error (TE) for evaluation. It measures the mean absolute error between point in the tail face for the prediction and ground truth of the workpiece.

Additionally, we use relative error reduction to compute the promotion portion w.r.t. the second best model, which is formulated as $\frac{\text{The second best error} - \text{Our error}}{\text{The second best error}} \times 100\%$.

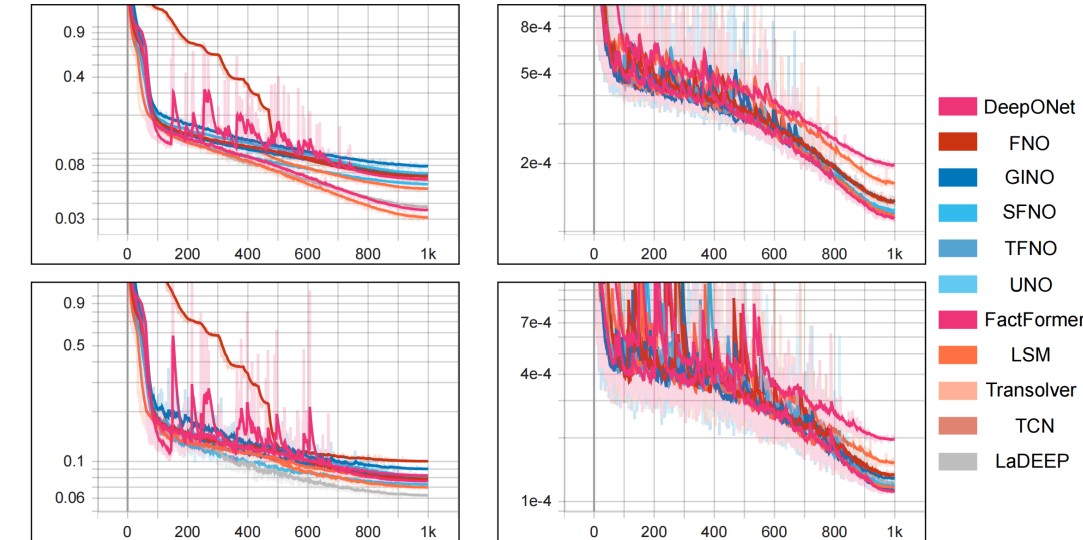

Figure 16: The training and evaluation loss curves of LaDEEP and baselines in setting (2). For all sub-figures, the x-axis is the epochs and the y-axis is the corresponding losses. From top to bottom, from left to right, they are: the training $loss_p$, the training $loss_r$, the evaluation $loss_p$, and the evaluation $loss_r$. The smooth parameter is 0.9.

### A.6    ANALYSIS FOR SAMPLES WITH LARGE ERRORS

Table 7 shows the maximal and minimal results of LaDEEP and baselines. For samples with large errors, LaDEEP does not perform the best. This indicates that there is still room for improvement in LaDEEP when handling samples with large errors.

We visualize two samples with large errors in Figure 17. We observe that errors are mainly due to the underestimation of the predictions along the z axis. In most cases, the z-direction (thickness direction of the workpiece) is the primary direction in which the workpiece is formed over the mold. When there is a large displacement along the z-axis, prediction becomes more challenging. This is actually a point worth paying attention to.

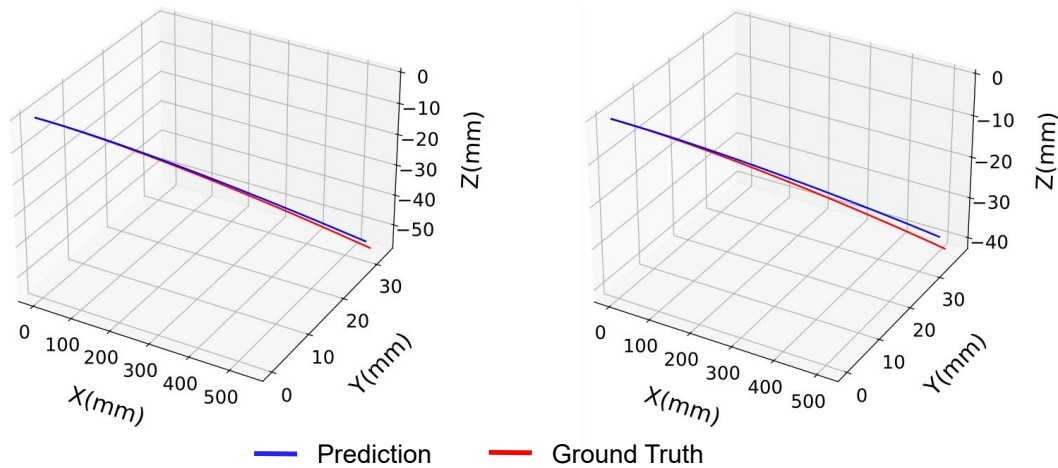

Figure 17: Visualization of two samples with large error in MAD (left: 1.48mm, right: 1.77mm).

Table 7: Best and worst samples of models in both settings described in Section 3.2. **Up**: Results in setting (1); **Down**: Results in setting (2). For MAD and TE, a smaller value indicates better performance, whereas for Iou 3D, the opposite is true.

| Model | MAD(mm) | | IoU 3D(%) | | TE(mm) | |
|---|---|---|---|---|---|---|
| | min | max | min | max | min | max |
| DeepONet | 0.0968 | 1.8079 | 24.71 | 99.49 | 0.0624 | 4.6450 |
| FNO | 0.0637 | 1.7474 | 24.88 | 97.44 | 0.0670 | 5.1231 |
| GINO | 0.0631 | 2.0506 | 24.23 | 96.30 | 0.0687 | 5.6545 |
| SFNO | 0.0779 | 1.8645 | 24.61 | 98.90 | 0.0713 | 5.3106 |
| TFNO | 0.0747 | 1.7692 | 24.71 | 98.10 | 0.0940 | 4.6956 |
| UNO | 0.0797 | 1.8011 | 22.43 | 97.26 | 0.0796 | 5.1126 |
| FactFormer | 0.0608 | 2.8023 | 14.24 | 98.42 | 0.0564 | 8.1669 |
| LSM | 0.0823 | 2.5066 | 20.19 | 97.38 | 0.0618 | 6.8324 |
| Transolver | 0.0807 | 2.0487 | 23.07 | 98.73 | 0.0894 | 5.6476 |
| TCN | 0.0733 | 1.8332 | 22.09 | 98.87 | 0.0803 | 4.8999 |
| LaDEEP(ours) | 0.0183 | 1.8435 | 24.39 | 99.30 | 0.0397 | 5.4094 |

| Model | MAD(mm) | | IoU 3D(%) | | TE(mm) | |
|---|---|---|---|---|---|---|
| | min | max | min | max | min | max |
| DeepONet | 0.0244 | 1.7669 | 26.22 | 99.41 | 0.0286 | 5.4313 |
| FNO | 0.0321 | 1.9502 | 24.62 | 98.74 | 0.0182 | 5.9502 |
| GINO | 0.0371 | 1.8564 | 24.00 | 99.61 | 0.0584 | 5.3937 |
| SFNO | 0.0324 | 2.1324 | 23.33 | 99.29 | 0.0536 | 6.2001 |
| TFNO | 0.0314 | 1.8642 | 24.80 | 98.95 | 0.0480 | 5.4331 |
| UNO | 0.0251 | 2.0244 | 22.87 | 98.86 | 0.0524 | 5.7634 |
| FactFormer | 0.0221 | 2.1131 | 23.04 | 99.26 | 0.0281 | 6.0887 |
| LSM | 0.0346 | 1.8776 | 25.37 | 99.82 | 0.0697 | 5.5618 |
| Transolver | 0.0311 | 1.9816 | 24.67 | 99.03 | 0.0227 | 5.7928 |
| TCN | 0.0184 | 2.0861 | 26.15 | 98.36 | 0.0499 | 6.0920 |
| LaDEEP(ours) | 0.0183 | 1.8435 | 24.39 | 99.30 | 0.0397 | 5.4094 |

## A.7 TRANSFERABILITY ON NEW MATERIALS

In metal fabrication, stretch bending is a key profile processing technique, widely applied in automotive manufacturing, aeronautical engineering, and similar fields. This process represents a typical challenge involving large elastic-plastic deformation. We see it as an ideal starting point for applying AI to problems in large elastic-plastic deformation. Additionally, we believe that the proposed framework, with its explicit integration of physical modeling, can be extended to related tasks such as stamping, forging, and more through transfer learning or customized feature extractors. Future work will focus on developing improved models and tackling more complex tasks. Additionally, to further explore the model's transfer ability, we conduct further experiments to assess its potentials in transferring to other tasks.

This experiment is to explore the model's transferability across different metal materials. We use a new aluminum alloy with different alloy ratios compared to the original dataset. These alloys differ in material parameters: hardness, density, Poisson's ratio, Young's modulus, and stress-strain behavior, leading to significant differences in deformation, stress, and rebound behavior. The parameters for both alloys are measured from materials used in practical production. Using the 5 cross-sections mentioned in the paper, we generate a total of 300 data samples, which are split into an 8:2 ratio for training and test. Two experiment settings: (1) We take the model in the paper as the basic pre-trained model with is trained on the original dataset for 600 epochs. Then we test the new data directly on the pre-trained model. (2) We fine-tune the pre-trained model in (1) for 200 epochs with the new dataset. The results are presented in Table 8.

Table 8: Results of Cross-section Generalization.

| Setting ID | MAD(mm) | IoU 3D(%) | TE(mm) |
|:----------:|:-------:|:---------:|:------:|
| 1 | 0.2053 | 84.43 | 0.5338 |
| 2 | 0.1711 | 85.56 | 0.4389 |

These results demonstrate that with our pre-trained model, we can achieve rapid convergence on new alloy data using only a small dataset and minimal fine-tuning iterations. This capability is highly valuable in practical applications where adapting to new materials efficiently is crucial. We acknowledge that these experiments only explore generalization to unseen cross-sections and transferability across different materials. While it does not fully demonstrate the model's generalization across various industrial scenarios and tasks, it highlights the potential of the proposed framework. We believe that with further structural improvements and the combined use of pre-training and fine-tuning techniques, this framework can be effectively extended to cover a broader range of applications.

### A.8 TWO-LOOP MOLD DESIGN PARADIGM

Errors in mold design process are mainly categorized into simulation error (SE), application error (AE) and production error (PE). The SE is the discrepancy between the result of simulation and the target shape, defined within the virtual space. However, there is often a gap between simulation results and their practical application due to various sources of factors arise at different stages of process. For example, mold production inherently involves certain tolerances which are introduced by the manufacturing machines. Installation tolerance arises when installing the mold onto the machine. The motion parameters of working arms may exhibit zero-point drift, and the properties of workpiece change over time, among other factors. The AE is defined as the error generated when simulation results are directly implemented in production, within 10mm. This gap is inevitable due to the inherent complexities. Then on-site adjustments will be carried out. On-site adjustment methods include machining of the mold by tools directly, as well as fine-tuning the parameters of working arms. By employing these methods, the final PE will be kept below 1mm for production.

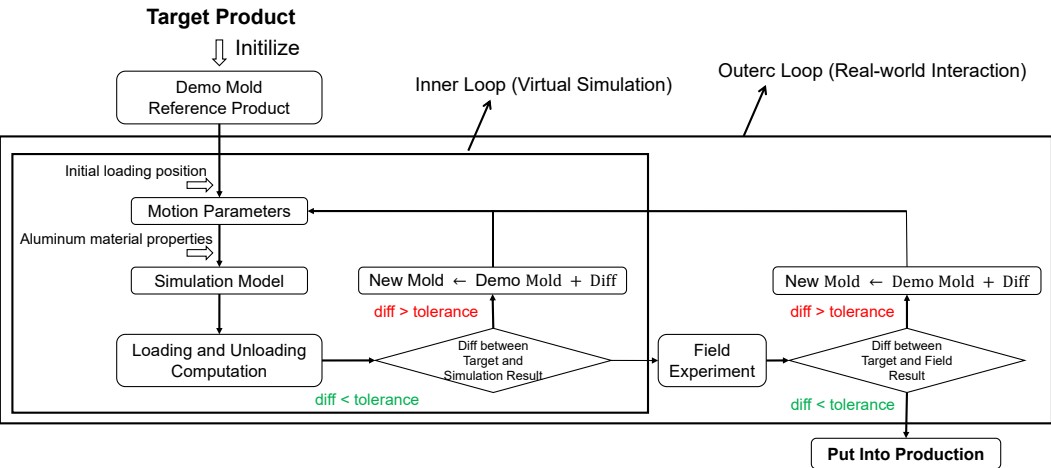

Figure 18: Two-loop mold design paradigm.

In proposed two-loop mold design paradigm in Figure 18, we use inner and outer loop to guarantee the reliability and meet all errors mentioned above. In inner loop, we use simulation and displacement compensation method (Cafuta et al., 2012) to iteratively obtain a simulation result meet the SE and AE. The simulation is FEM in traditional and LaDEEP now. Then through on-site adjustment, PE could be met in most cases. If, after several adjustments, the on-site results still fail to meet production standard, it is necessary to feedback the actual outcomes to the simulation system. This feedback process, called outer loop, is crucial as it is a recalibration for the compensation. And it ensures a seamless transition from the virtual simulation space to the real-world production environment.

We start from a naïve design of mold (say a straight one), simulate the deformation process, compute the distance between the simulation results and the desired shape. This distance helps us to re-design the mold, and the computations start over. This cycle is conducted purely in computations, and terminates until the workpiece deforms as designed after rebound. The mold will be delivered to the factory and produced in real world. There will also be errors after real-world production, as there are errors on simulation, which will be feedback to our LaDEEP model for further design.

A.9    VISUALIZATIONS

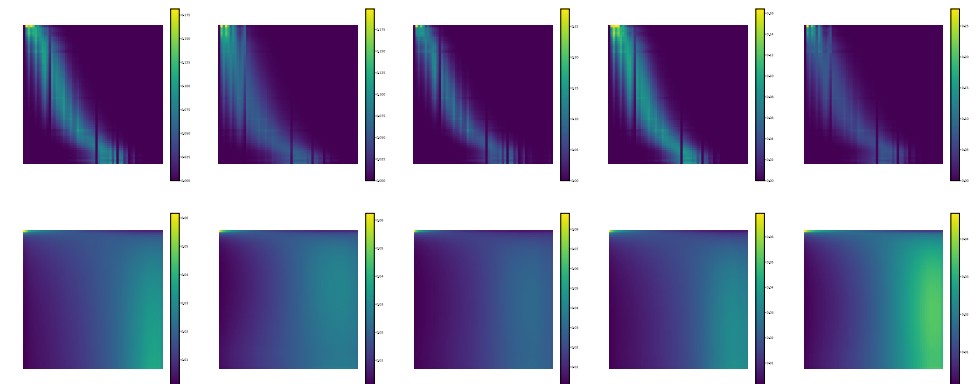

Figure 19: Visualization of attention maps. The patterns described in Sec 2.3 universally exist.

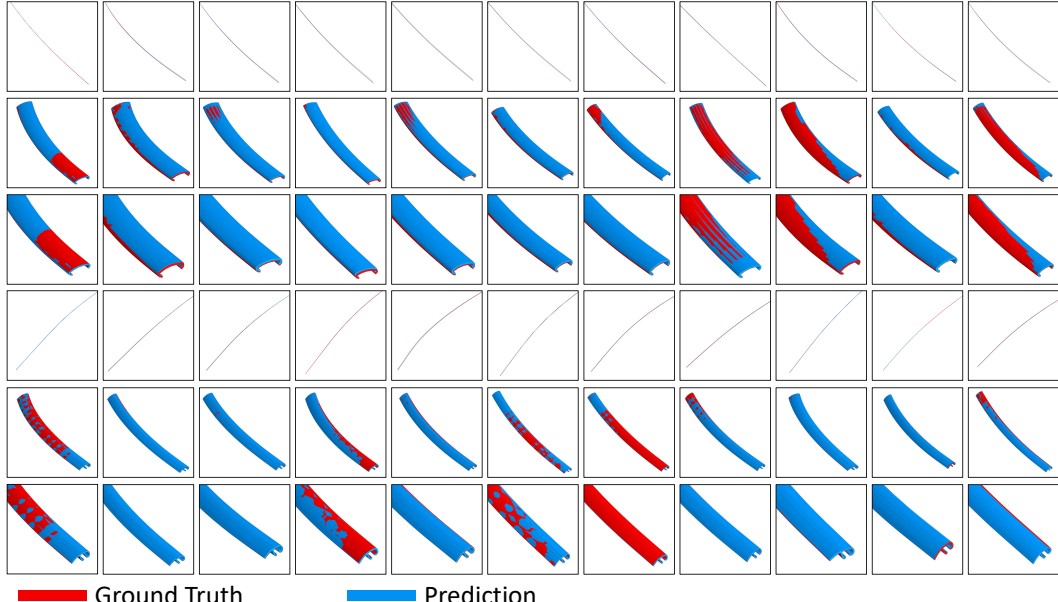

Figure 20: Visualization of results. Columns from left to right correspond to LaDEEP, DeepONet, FNO, GINO, SFNO, TFNO, UNO, FactFormer, LSM, Transolver and TCN, respectively. Rows 1-3 represent a sample with a type-3 cross-section. Rows 4-6 represent a sample with a type-5 cross-section. Rows 1 and 4 are tails of the characteristic lines. Rows 2 and 5 are global views of the workpieces. Rows 3 and 6 are tails of the workpieces.

