# OpenReview forum: "LaDEEP: A Deep Learning-based Surrogate Model for Large Deformations of Elastic-Plastic Solids"
_ICLR.cc/2025/Conference — ICLR 2025 Conference Withdrawn Submission_

### Official Review · Reviewer_BH3A · 2024-10-28

**Soundness:** 2
**Presentation:** 1
**Contribution:** 2
**Rating:** 3
**Confidence:** 4

**Summary:**

This work proposes a transformer based deep-learning model to learn elasto-plasticity effects on solid objects with linear (slender) cross-sections. The model achieves 5x speedup compared to FEM solvers and an average improvement of 20.5% on other deep-learning based solvers. The proposed architecture includes a 2-stage transformer based module. A key contribution of the paper is practical implementation of the model, put into real production.

**Strengths:**

Originality: A key contribution of the paper is the practical implementation of the work in production settings, which showcases the robustness of the proposed architecture in real-world settings, as opposed to toy-problems that a lot of ML papers showcase.

Quality: The motivation of the paper is clearly stated. The architecture is described thoroughly, and is presented clearly. There are several deep-learning baselines that the model is compared against.

Significance: This work has applications in engineering and scientific design, providing a faster and reasonably accurate surrogate model to learn elastoplasticity.

**Weaknesses:**

The authors claim that they are the first deep-learning based architecture to model elastic and plastic behavior. This may be a misleading claim. There are several works that attempt to model plastic and elastic behavior in different physical systems. While they may not be doing exactly what this work proposes, there exist several deep-learning models in the field regardless. It might help to differentiate the research gaps between the model proposed by the authors and the following works

1.	Lechner, Philipp, et al. "A physically-informed machine learning model for freeform bending." Journal of Intelligent Manufacturing (2024): 1-13.
2. Zong, Zeshun, et al. "Neural stress fields for reduced-order elastoplasticity and fracture." SIGGRAPH Asia 2023 Conference Papers. 2023.
3.	Li, Xuan, et al. "Pac-nerf: Physics augmented continuum neural radiance fields for geometry-agnostic system identification." arXiv preprint arXiv:2303.05512 (2023).
4.	He, Junyan, et al. "A deep learning energy-based method for classical elastoplasticity." International Journal of Plasticity 162 (2023): 103531.
5.	Kim, Suhan, and Hyunseong Shin. "Deep learning framework for multiscale finite element analysis based on data-driven mechanics and data augmentation." Computer Methods in Applied Mechanics and Engineering 414 (2023): 116131.
7.	Xiong, Zhihao, Ping Yang, and Pengyang Zhao. "Physics-driven neural networks for nonlinear micromechanics." International Journal of Mechanical Sciences 273 (2024): 109214.
8.	Yang, Zhenze, Chi-Hua Yu, and Markus J. Buehler. "Deep learning model to predict complex stress and strain fields in hierarchical composites." Science Advances 7.15 (2021): eabd7416.

The authors claim their model can learn complex and large deformations but seem to show a single setting (stretch bending) on a single dataset. It would be good to showcase the performance against more datasets.

The quality and placement of figures can be greatly improved. Figure 2.b for instance should be placed closer to results section rather than the introduction section. Figure 1 is contained within Figure 3.

Some of the elements of the architecture, which are used from other works, are explained in too much detail.  For instance, Equation 2, is part of the transformer architecture, adapted from Vaswani et al., 2017.

**Questions:**

1.	As mentioned in the weakness section, there are several deep-learning based models for learning elasto-plastic behavior, both in fluids and solids. I would like to know how this work addresses the research gap from the above mentioned works. Perhaps the literature review should include deep-learning based elasto-plasticity models.

2.	The metrics proposed in the paper are specific to this application. However, models such as FNO, DeepONet, etc. are general purpose Deep Learning models for learning PDEs. It would help to show the Relative error (L2) between the predicted results and the ground truth to better understand performance improvements.

3.	In Figure 7, it seems like LSM is visually indistinguishable from LaDeep. However the numbers presented in Table 1 seem to indicate otherwise.

---

### Official Review · Reviewer_QUVT · 2024-11-01

**Soundness:** 3
**Presentation:** 3
**Contribution:** 2
**Rating:** 3
**Confidence:** 5

**Summary:**

This paper presents a modified transformer architecture for large-scale but specific elastic-plastic solid simulation. The technical execution is sound, with clear writing and well-reasoned adaptations, eg two-stage transformer blocks for loading and unloading respectively, to address specific challenges in their elastic-plastic simulation.

However, there are fundamental concerns about the paper's scope and broader impact as a ML conference paper:
1. Limited Novel Machine Learning Contributions
    - The core approach relies heavily on established transformer architectures;
    - Modifications, while appropriate to their specific case, are primarily engineering adaptations and not generizable to general elastic-plastic simulations (as the general case does not necessarily follow a loading-unloading procedure)
    - It's reasonable to assume if the loading pattern changes, the proposed specific architecture will not general well

2. Limited Impact on ML Research Community (Even if we narrow down into a few sub-domains, ML4Sci, ML4PDE, or ML4Mechanics; due to the limitation of the loading pattern mentioned before)

3. To my knowledge, this is also not the 1st paper that tries to leverage ML in plasicity simulation. The authors should consider the following citation, and read through its literature review, which is quite comprehensive in my mind. Adding these more related works will also increase the quality of manuscript when authors try other domain specific journals:
Plasticitynet: Learning to simulate metal, sand, and snow for optimization time integration

While technically sound, I recommend rejection for an ML conference venue. The paper would be better suited for domain-specific journals such as:

- Computational Mechanics
- Journal of Computational Physics
- Computer Methods in Applied Mechanics and Engineering

**Strengths:**

- Clear writing
- Technical details are sound and presented well
- The modification such as using image to represent the loading characteristic line, and using 2 blocks for loading and unloading, respectively are reasonable
- The results and real experiments are good

**Weaknesses:**

- See overview; main drawback is the scope is limited, even if we only consider a sub-domain, such as ML4PDE, ML4Mechanics, due to the too specific architecture and application case.

**Questions:**

- NA

---

### Official Review · Reviewer_C3eT · 2024-11-01

**Soundness:** 2
**Presentation:** 2
**Contribution:** 1
**Rating:** 1
**Confidence:** 5

**Summary:**

The paper introduces LaDEEP, a deep learning-based model for predicting large deformations of elastic-plastic solids, specifically focusing on the industrial process of stretch bending. LaDEEP employs a Transformer-based architecture, the Deformation Predictor (DP), which captures complex interactions in loading and unloading processes using property-aware token sequences. The model encodes 3D solids into a structured format, leveraging cross-attention and self-attention mechanisms to accurately predict the final shape of a deformed workpiece. Finally, the authors claim that LaDEEP achieves up to five magnitudes faster computation speed than FEM with comparable accuracy while exceeding the accuracy of state-of-the-art and has been deployed in real-world industrial settings.

**Strengths:**

The application of LaDEEP to the problem of large elastic-plastic deformations is both innovative and compelling. The real-world deployment underscores its practical impact, showcasing the model’s potential to significantly accelerate computations while maintaining high accuracy. This advancement holds substantial implications for industrial applications, especially in manufacturing processes that rely on iterative simulations. The research is well-structured, containing all the essential components of a strong study, and the presentation is effective. The use of figures to illustrate the stretch bending process and attention mechanisms enhances the clarity of the complex methodology.

**Weaknesses:**

The authors suggest that existing models are not well-suited for the specific task and have thus developed their own simulated dataset. While this approach is understandable, it may introduce a risk of bias, as the model's performance could appear stronger when tested on data specifically tailored to its design. To enhance the robustness of their comparisons, incorporating additional standardized datasets, such as FNO [1] or geo-FNO [2], would strengthen the validity of their results.

The presentation of the model architecture could be clearer. Currently, the explanation is challenging to follow, and the novelty in model design appears limited, as it primarily adapts existing Transformer techniques to a new context rather than introducing a fundamentally unique approach. The visuals, while helpful, could be improved by diversifying the images to avoid repetition. Additionally, simplifying the physical problem by assuming symmetry and simulating only half of the workpiece, while practical, may limit the model's applicability. Including simulations of asymmetric systems could provide a more comprehensive demonstration of the model’s capabilities. Lastly, incorporating a discussion on critical factors such as surface roughness and the stress-strain relationship would further bolster the study’s relevance and completeness.

[1] Li, Z., Kovachki, N., Azizzadenesheli, K., Liu, B., Bhattacharya, K., Stuart, A., & Anandkumar, A. (2020). Fourier neural operator for parametric partial differential equations. arXiv preprint arXiv:2010.08895.

[2] Li, Z., Kovachki, N., Choy, C., Li, B., Kossaifi, J., Otta, S., ... & Anandkumar, A. (2024). Geometry-informed neural operator for large-scale 3d pdes. Advances in Neural Information Processing Systems, 36.

**Questions:**

1) How do you address concerns that your in-house generated dataset may bias the results in favor of LaDEEP, potentially limiting the validity of comparisons with state-of-the-art methods? Have you considered using an external or more standardized dataset to strengthen the evaluation?

2) Could you provide more detailed information about the data generation process, such as mesh resolution, material parameters, or simulation conditions in Abaqus? This would improve the reproducibility and credibility of your dataset.

3) The current presentation of LaDEEP’s model architecture is challenging to follow. Have you thought about simplifying or restructuring this section for better clarity, perhaps with more intuitive diagrams or a step-by-step explanation? Moving figures like 10 or 11 from the supplementary material to the main text could also improve comprehension.

4) Given that LaDEEP primarily applies a Transformer-based architecture to a physical simulation problem, what do you consider the most significant model-wise innovation? How does your approach differ fundamentally from existing transformer applications?

5) The model simplifies the problem by assuming symmetry and only simulating half of the workpiece. Have you considered including simulations of asymmetric systems in your dataset to provide a more comprehensive validation of LaDEEP? How might this impact your results and conclusions?

6) Surface roughness is an important factor in deformation [3, 4]. How does LaDEEP account for surface roughness, if at all? If this factor was not considered, do you see incorporating it as a potential area for future improvement, and how might it affect your model’s accuracy?

7) The stress-strain relationship is crucial for predicting elastic-plastic deformations [5, 6]. Could you elaborate on how your model accounts for or adapts to different stress-strain curves, and whether any material-specific calibration is necessary? If not, do you see this as a limitation that could affect the model's generalizability?


[3] Tiwari, A., Almqvist, A., & Persson, B. N. J. (2020). Plastic deformation of rough metallic surfaces. Tribology Letters, 68(4), 129.

[4] Persson, B. N. J. (2023). Surface Roughness-Induced Stress Concentration. Tribology Letters, 71(2), 66.

[5] Huda, Z., & Huda, Z. (2022). Stress-Strain Relations and Deformation Models. Mechanical Behavior of Materials: Fundamentals, Analysis, and Calculations, 109-118.

[6] Bertram, A. (2012). Elasticity and plasticity of large deformations. Berlin: springer.

---

### Official Review · Reviewer_BoUX · 2024-11-03

**Soundness:** 2
**Presentation:** 3
**Contribution:** 2
**Rating:** 3
**Confidence:** 3

**Summary:**

The paper presents a framework for learning large deformations of elastic-plastic solids. The setting of the model is restricted to stretch bending of slender structures with self-similar cross-section. The model predicts the deflection of the characteristic line after loading and unloading (with the associated rebound).

**Strengths:**

The framework presents a novel use of transformers in the domain of structural/solid mechanics. The authors have empirically shown that the framework can be useful in a two-loop optimization scenario when the mold shape needs to be designed in production setting.

The model is benchmarked against many other ML models.

The composition of the model is complex and sound, with several good choices - e.g. the use of decoders to force a meaningful encoding of the cross-section.

**Weaknesses:**

Despite the fact that the authors showed practical deployment of the model, there are numerous areas which are holding the paper back from being a significant contribution in the ML community. I discuss them in two broad areas here, while the “Questions” part will have further specific prompts.

**1.	Only very limited generalizability was demonstrated**

Much of the paper (including Table 1) is focused on both training and evaluating the model on the same “in-distribution” dataset which included 600 samples of each of the five types of cross-section. At that point, it is basically just interpolation with no generalization capability.

In cases where the authors do show “OOD” tests (Transferability in 3.2, zero-shot), the gap becomes obvious – the attempt to generalize from sections 1-4 to section 5 results in a much larger error. Furthermore, the difference between section 5 and the other 4 sections is not very large. What the authors call “fine-tuning” is done on 480 new samples of X-section 5 (out of 600 total samples). This is a huge size of the additional dataset that is brought in. Recall that FEM takes a lot of time. How much time does it take to collect the data for “fine-tuning” on 480 additional samples? How would the results change if the fine-tuning was done on a more sensible number of new samples (e.g. 40)?

Additional experiments on generalization should be done. For instance, what happens if you test the model on an example where the initial characteristic line in curved? Perform more rigorous transferability test – e.g. train on X-sections 1-3 and test on 4. Or make another testing X-section with an additional horizontal segment - i.e. sections 2-3 have two horizontal segments while section 4 has three; can you test on a section with four?

**2.	Solid mechanics insights are missing**

The only comparison of predictions with respect to the ground truth are the deflection of the characteristic line. While I acknowledge that that is the main desired output of the model, it would be instructive to discuss how the model might be violating the physical laws locally, Especially in the cases with high error (e.g. A.6) – to what extent are laws such as momentum balance $(\nabla.\sigma=0)$ violated?

A big aspect of the real physics that the model ignores is that the cross-section can change shape during deformation (e.g. warping). To what extent can this be important in accuracy of predictions? Moreover, there is little discussion of aspects of contact mechanics – e.g. friction.

**Questions:**

The following typos were found:

-	15: “handle ~~with~~ complex”
-	195: “the middle of the workpiece _is_ kept static”
-	279: grammar of the sentence “Recall that…”
-	302: missing comma between Linear
-	Many instances of units in italics (e.g. 378-388). Use \mathrm{mm} or \SI{0.18}{mm}
-	435: grammar of the sentence “However, the considered…”
-	533: “existing models”
-	533: grammar of the sentence “However, due to…”

Many times numbers are quoted to four significant figures. This is meaningless, unless the error bars on these quantities are that small. For instance, how many experiments are run to obtain each number in Table 1? Do you really have a sample size big enough to quote 4 significant figures? Similarly when using “20.47%” in abstract.

In caption of Figure 2, please refer the reader to the definitions of the abbreviations (MAD).

134: The criticism of PINNs that “they require the precise formula of PDEs” is only partially justified. PINNs were shown to be able to infer the coefficients of PDEs.

Why do we observe vertical bands of low weights in Fig. 5b?

285: define LN – presumably LayerNorm?

The model is not mesh-free. For FEM, various mesh resolutions are shown. Please also show the results for various discretizations for your model.

Please show convergence studies: (1) how test error reduces with an increasing number of samples used for fine-tuning; (2) convergence with the number of examples in the dataset. Do you need 600 training samples? How much worse is the model when trained with 100?

Please give estimates of how much time approximately it takes to run FEM for, say 600 samples.

---

### Note · Authors · 2024-11-20

I have read and agree with the venue's withdrawal policy on behalf of myself and my co-authors.